# Loxl2 and Loxl3 Paralogues Play Redundant Roles during Mouse Development

**DOI:** 10.3390/ijms23105730

**Published:** 2022-05-20

**Authors:** Patricia G. Santamaría, Pierre Dubus, José Bustos-Tauler, Alfredo Floristán, Alberto Vázquez-Naharro, Saleta Morales, Amparo Cano, Francisco Portillo

**Affiliations:** 1Departamento de Bioquímica, Instituto de Investigaciones Biomédicas Alberto Sols, Universidad Autónoma de Madrid, CSIC-UAM, 28029 Madrid, Spain; jbtauler@gmail.com (J.B.-T.); alfredo.floristan@gmail.com (A.F.); avazquez@iib.uam.es (A.V.-N.); smorales@iib.uam.es (S.M.); amparo.cano@inv.uam.es (A.C.); 2Instituto de Investigación Sanitaria del Hospital Universitario La Paz-IdiPAZ, 28029 Madrid, Spain; 3Centro de Investigación Biomédica en Red, Área de Cáncer (CIBERONC), Instituto de Salud Carlos III, 28029 Madrid, Spain; 4Faculty of Medicine, Université de Bordeaux, INSERM U1312 BRIC, BoRdeaux Institute in onCology, 33076 Bordeaux, France; pierre.dubus@u-bordeaux.fr; 5CHU de Bordeaux, Institute of Pathology and Cancer Biology, Haut-Lévêque Hospital, 33600 Pessac, France

**Keywords:** lysyl oxidases, Loxl2, Loxl3, epistasis analysis, embryonic lethality

## Abstract

Lysyl oxidase-like 2 (LOXL2) and 3 (LOXL3) are members of the lysyl oxidase family of enzymes involved in the maturation of the extracellular matrix. Both enzymes share a highly conserved catalytic domain, but it is unclear whether they perform redundant functions in vivo. In this study, we show that mice lacking Loxl3 exhibit perinatal lethality and abnormal skeletal development. Additionally, analysis of the genotype of embryos carrying double knockout of *Loxl2* and *Loxl3* genes suggests that both enzymes have overlapping functions during mouse development. Furthermore, we also show that ubiquitous expression of Loxl2 suppresses the lethality associated with Loxl3 knockout mice.

## 1. Introduction

The lysyl oxidase (LOX) family is composed of five lysine-tyrosylquinone (LTQ)-dependent copper amine oxidases: LOX and four LOX-like paralogs LOXL1–4. All members are characterised by a highly conserved carboxyl (C)-terminal amine oxidase catalytic domain. This domain includes a histidine-rich copper-binding motif and a lysyl–tyrosyl–quinone (LTQ) cofactor, both essential for catalytic activity, and a cytokine receptor-like (CRL) domain whose function remains unknown [1,2,3,4]. By contrast, the amino (N)-terminal region diverges among all members. In the case of LOXL2–4, it presents four scavenger receptor cysteine-rich (SRCR) domains whose functional role has not been well characterised yet, although they could be involved in protein–protein interactions [5]. Based on N-domain diversification and sequences comparisons, these proteins have been classified into two subfamilies: one constituted by LOX and LOXL1, while LOXL2, LOXL3, and LOXL4 belong to the second one [2,4]. The physiological function of LOX enzymes is the maturation of the extracellular matrix (ECM). Lysyl oxidases catalyse the oxidative deamination of ε-amino groups of peptidyl–lysine and hydroxylysine residues to produce highly reactive aldehydes—allysine residues—that undergo a spontaneous condensation, thus establishing intra- or inter-cross-linkages in collagen and elastin [4,6]. Besides their critical role in ECM maturation, lysyl oxidase proteins are also associated with diverse pathologies, including fibrosis, cancer, and cardiovascular diseases (reviewed in [7,8,9,10,11,12,13]).

The generation and characterisation of genetically modified mouse models with gain or loss of function of Lox [14,15,16], Loxl1 [17,18], Loxl2 [19,20], Loxl3 [21,22] and Loxl4 [23] have highlighted the critical role of LOX enzymes in mammalian development. The genetic ablation of *Loxl2* or *Loxl3* leads to perinatal lethality with incomplete penetrance caused by different alterations. *Loxl2* KO mice lethality is associated with congenital heart defects and/or distension of the hepatic blood vessels [19], while *Loxl3* KO mice lethality has been linked to impaired embryonic development of the palate shelves, vertebral column and pulmonary system [21,24]. Moreover, Loxl3 was shown to be key for proper muscle development [22]. These different phenotypes indicate that there is no functional compensation between both enzymes during embryonic development despite presenting 71% of identity in the catalytic domain [3]. Nevertheless, this assumption has not been demonstrated to date. Therefore, we decided to study the functional relationship of Loxl2 and Loxl3 using in vivo models. To this end, we generated double transgenic mice carrying either double *Loxl2* and *Loxl3* knockout (KO) genes or a *Loxl2* knockin (KI) gene in a *Loxl3* KO background. We have observed that double *Loxl2*/*Loxl3* KO results in embryonic lethality, whereas the ubiquitous expression of *Loxl2* can suppress the perinatal lethality associated with *Loxl3* KO mice.

## 2. Results

### 2.1. Deletion of Loxl3 Provokes Perinatal Lethality

We first confirmed the described perinatal lethality of *Loxl3* KO mice [21,22]. To this end, we generated heterozygous *Loxl3*^+/*LacZ*^ mice as recently described [25] and analysed the lethality associated with the *Loxl3^LacZ^*^/*LacZ*^ genotype upon breeding heterozygous mice. The quantification of the offspring after weaning revealed that the percentage of *Loxl3^LacZ^*^/*LacZ*^ mice was 5.64%, a five-fold lower frequency than expected from a Mendelian ratio (Table 1). Since at E18.5 the expected number of *Loxl3^LacZ^*^/*LacZ*^ embryos was observed (Table 1), we concluded that *Loxl3* KO mice died perinatally. Of note, the percentage of *Loxl2*^−/−^ surviving animals after weaning is 10.7%, two-fold lower than expected [19].

### 2.2. Loxl3 KO Mice Present Skeletal Abnormalities

We then analysed the phenotype of *Loxl3* KO surviving mice. *Loxl3^LacZ^*^/*LacZ*^ mice showed significantly smaller size, consistent with lower weight and shorter length of tibia and femur (Figure 1A,B), and a poor health appearance that resulted in a higher mortality rate than their corresponding littermates, determined by closely observing *Loxl3*^+/+^ (n = 26) and *Loxl3^LacZ^*^/*LacZ*^ (n = 61) mice for more than a year. In addition, most *Loxl3^LacZ^*^/*LacZ*^ adult animals presented alterations in locomotion, imbalance and spinal deformities (Appendix A), the latter observed by Zhang et al. in newborn *Loxl3* KO mice [21]. However, in *Loxl3^LacZ^*^/*LacZ*^ mice, we did not detect any of the craniofacial abnormalities described previously [21], suggesting that the perinatal lethality we observed was not due to cleft palate.

To investigate the causes of the *Loxl3* KO phenotype, we first analysed the expression of *Loxl3* in *Loxl3*^+/*LacZ*^ embryos at E12.5, 14.5 and 16.5 developmental stages. To this end, we took advantage of the *Loxl3^LacZ^* allele that expresses the *LacZ* gene under the control of the *Loxl3* promoter [25]. In this way, it was possible to analyse the expression of *Loxl3* during embryonic development by X-Gal staining. As can be seen in Figure 2A, we detected the expression of *Loxl3* at E12.5 in various regions of the head (chondrocranium and cochlear area), fourth ventricle of the choroid plexus, axial zone, kidney, heart, and faint expression in the hindlimbs. At E14.5 and E16.5 stages, the expression of *Loxl3* is comparable and more spatially defined, showing an intense staining in pre-osseous cartilaginous structures such as cartilage precursors of the occipital and interparietal bones, nasal area, scapula and costal cartilages, as well as in the epiphysis of long bones, hip or humeral head, elbow, wrist and slight expression in vertebrae (Figure 2A). These data indicate that Loxl3 may have a role in mouse skeletal development.

To gain insight into this possible function of Loxl3, littermate E18.5 embryos (n = 4 WT and n = 4 KO) were stained with alcian blue/alizarin red, a technique that distinguishes cartilage from mineralised bone (Figure 2B). The measurement of the length of the different ossified skeletal structures from the embryos showed a significant difference in the global relative length of these ossified elements in *Loxl3^LacZ^*^/*LacZ*^ embryos compared to wild-type controls (Figure 2C).

To analyse the observed defects in postnatal skeletal development, paraffin sections from the femur of one-month-old *Loxl3*^+/+^ and *Loxl3^LacZ^*^/*LacZ*^ littermates were stained with haematoxylin/eosin. Histopathological analysis showed that some *Loxl3^LacZ^*^/*LacZ*^ mice exhibit dysplasia in the upper femoral epiphysis, with atrophy of the head of the femur, hypertrophy of the greater trochanter and an abnormal angle arrangement of these two structures (Figure 3A). This could contribute to the locomotor problems that we observed in some of the *Loxl3* KO mice (Appendix A). Additionally, the mature lamellar bone of the diaphysis of *Loxl3^LacZ^*^/*LacZ*^ mice is interrupted by zones of immature reticular embryonic bone, with osteoblasts and chondrocytes inside (Figure 3B). In certain areas, the diaphysis compact bone is replaced by cancellous bone (Figure 3C). This anomalous organisation would be responsible for the greater skeletal fragility detected de visu when processing the bones of animals that lacked Loxl3.

Altogether, our results suggest that Loxl3 contributes to skeletal development and that aberrant bone maturation might be the underlying cause or significantly contribute to the perinatal lethality observed in *Loxl3* KO mice.

### 2.3. Double Knockout of Loxl2 and Loxl3 Genes Leads to Embryonic Lethality

We then proceeded to analyse the viability of double *Loxl2*/*Loxl3* KO (*Loxl2*^−/−^; *Loxl3^LacZ^*^/*LacZ*^) mice by breeding *Loxl2*^+/−^ [19] and *Loxl3*^+/*LacZ*^ heterozygous mice. We could not find any double *Loxl2*/*Loxl3* KO mouse among the offspring from the crossings (n = 293) (Table 2).

To discern whether the observed lethality in the double KO mice occurred during embryogenesis or postnatally, we harvested embryos at E9.5, E11.5 and E13.5 developmental stages. Genotype analysis indicated that *Loxl2*^−/−^; *Loxl3^LacZ^*^/*LacZ*^ embryos appeared with a normal Mendelian frequency until E9.5, but not a single double *Loxl2*/*Loxl3* KO was observed at either E11.5 and E13.5 stages (Table 3). These data suggest that Loxl2 and Loxl3 are dispensable for early embryonic development up to E9.5 but are required for normal development thereafter.

### 2.4. Ubiquitous Expression of Loxl2 Ameliorates Perinatal Lethality in Loxl3 KO Mice

We previously developed a mouse model ubiquitously expressing *Loxl2* from the *ROSA26* locus (*R26^Loxl2^*^/+^) [19] that now allow us to analyse if *Loxl2* overexpression could rescue the *Loxl3* KO perinatal phenotype. To that end, we generated double heterozygous *R26^Loxl2^*^/+^; *Loxl3*^+/*LacZ*^ mice, bred them and analysed the offspring at weaning, around 21 days postnatally (n = 150). The expected Mendelian ratios of *Loxl3* KO mice expressing *Loxl2* ubiquitously were 6.25% for *R26^Loxl2^*^/*Loxl2*^; *Loxl3^LacZ^*^/*LacZ*^ and 12.5% for *R26^Loxl2^*^/+^; *Loxl3^LacZ^*^/*LacZ*^ mice (Table 4). The observed percentage of mice with the latter genotype fitted with the expected ratio, whereas less (2%) but still some *R26^Loxl2^*^/*Loxl2*^; *Loxl3^LacZ^*^/*LacZ*^ mice survived up to adulthood. No *Loxl3* KO mouse that was not overexpressing Loxl2 (*R26*^+/+^; *Loxl3^LacZ^*^/*LacZ*^) was found (Table 4), although the embryonic lethality in our single *Loxl3* KO model was not completely penetrant. These results suggest that ubiquitous Loxl2 expression rescues or strongly alleviates *Loxl3* KO-associated lethality. Only three *R26^Loxl2^*^/*Loxl2*^; *Loxl3^LacZ^*^/*LacZ*^ mice compared to 19 mice with the *R26^Loxl2^*^/+^; *Loxl3^LacZ^*^/*LacZ*^ genotype were found, suggesting that increased Loxl2 protein levels in *R26^Loxl2^*^/*Loxl2*^ compared to *R26^Loxl2^*^/+^ background [19] have a deleterious effect in mice lacking one or both wild-type *Loxl3* alleles (Table 4).

## 3. Discussion

The perinatal lethality shown by *Loxl3^LacZ^*^/*LacZ*^ mice as well as by other *Loxl3* KO models [21,22] points to a key role for Loxl3 in embryonic development. The embryonic expression pattern of *Loxl3* and the phenotypes associated with its genetic inactivation indicate that the main role of Loxl3 is essentially linked to ECM maturation and tissue homeostasis. Although the study by Zhang and collaborators [21] attributed the complete lethality they observed by P1 to the critical role of Loxl3 in palatogenesis, our *Loxl3* KO mouse model, in which we were not able to detect craniofacial defects, showed impaired skeletal development as indicated previously [21,22]. These differences, probably due to their different genetic backgrounds, allowed us to characterise *Loxl3* KO adult mice, which presented smaller size as well as locomotive, balance and skeletal abnormalities, resulting in a markedly shorter lifespan than *Loxl3* wild-type mice. Our data indicate that Loxl3 might have a role in endochondral ossification, plausibly linked to an altered ECM maturation provoked by the absence of Loxl3 during embryogenesis. We have shown that *Loxl3* is expressed in pre-ossified cartilage structures at least from E12.5, being detected in the long bones, mostly in the epiphysis, at E14.5 and E16.5. Defects in cartilage maturation would explain the cleft palate and spinal deformity observed in *Loxl3* KO mice [21] and in a zebrafish model lacking a LOXL3 orthologue [26]. Furthermore, a recent study using a targeted *Loxl3* deletion has demonstrated the relevance of Loxl3 in inner ear function, related to its role in type II collagen crosslinking [27] and required for the functional structure provided by the ECM. Human *LOXL3* has been proposed as a candidate gene responsible for recessive autosomal Stickler syndrome [28], a collagenopathy characterised by bone and cartilage abnormalities as well as by different degrees of hearing impairment [29] that might also cause imbalance, similar to the instability we observed in *Loxl3* KO adult mice.

Deletion of *Loxl2* causes incomplete perinatal lethality due to heart failure [19], whereas *Lox*-targeted mice died after birth due to cardiovascular instability [15]. The neonatal lethality we observed in our *Loxl3* KO mice might be related to cardiac defects as well as impaired lung development [24], which may contribute to death due to the physical stresses associated with parturition. Independent gene disruption of *Lox*, *Loxl2* and *Loxl3* results in lethality, indicating that the presence of other lysyl oxidases cannot compensate for their individual loss. These facts, together with the different phenotypes caused by the constitutive abrogation of Loxl2 or Loxl3, led to presume that no functional overlap exists between both enzymes, even if their catalytic domains present 71% identity. Nevertheless, this assumption has never been proven in in vivo models. We approached this question experimentally in two ways. First, we performed an epistasis analysis of *Loxl2* and *Loxl3* deletions in the mouse and observed that double *Loxl2* and *Loxl3* KO embryos die between E9.5 and E11.5 developmental stages, in contrast to the perinatal lethality observed in the independent single *Loxl2* [19] or *Loxl3* KO mutants. This result supports the idea that Loxl2 and Loxl3 functions may overlap to some degree in normal development and that they must operate in a common biochemical pathway required for embryonic viability after E9.5. Second, a rescue experiment of *Loxl3^LacZ^*^/*LacZ*^ perinatal lethality by ubiquitous expression of *Loxl2* was performed. The results showed that *Loxl2* ubiquitous expression ameliorates the perinatal lethality associated with *Loxl3^LacZ^*^/*LacZ*^ genotype, suggesting that the lack of functional complementation during embryonic development is not due to alternative functions of Loxl2 and Loxl3 proteins but either to their limiting expression levels or differential temporal and/or tissue distribution during development.

LOXL3, to the best of our knowledge, is the only lysyl oxidase with a crucial role in skeletal development. Further analyses of adult *Loxl3* KO mice expressing Loxl2 ubiquitously would be required to determine whether Loxl2 rescues the skeletal abnormalities associated with Loxl3 deficiency.

## 4. Materials and Methods

### 4.1. Mice

All mouse studies were performed in accordance with protocols approved by the Universidad Autónoma de Madrid Ethics Committee (ref # CEI-25-587) and the Comunidad de Madrid (PROEX 122/17). Mice were bred in a mixed genetic background (C57BL/6, CD1 and 129v strains). *Loxl2* constitutive KO mice (*Loxl2*^−/−^) and mice expressing *Loxl2* ubiquitously (*R26^Loxl2^*^/*Loxl2*^) were generated and genotyped as previously described [19]. To generate *Loxl3* constitutive KO mice (*Loxl3^LacZ^*^/*LacZ*^), embryonic stem cell clones bearing a knockout first allele (*Loxl3tm1a^(EUCOMM)Wtsi^*) were obtained from EUCOMM and were genotyped as described in [25]. Mouse strains have been deposited in the European Mouse Mutant Archive (EMMA) with accession numbers EM12882, EM13122 and EM13124.

### 4.2. Histology

Bones were dissected, cleared from muscle tissue and fixed in 4% formaldehyde overnight at 4 °C. Samples were decalcified with 0.5 M EDTA pH 8.0 for 48 h before embedding in paraffin. Five-micrometre thick sections were stained with haematoxylin and eosin solutions for morphological analysis.

### 4.3. Whole-Mount X-Gal Staining of Embryos

Mouse embryos at the indicated stages were collected in ice-cold phosphate-buffered saline (PBS) in 2 mL tubes. PBS was replaced by freshly prepared fixative solution (1% formaldehyde, 0.2% glutaraldehyde, 0.02% NP40 and 0.01% sodium deoxycholate in PBS) for 30 min at 4 °C on a rocking platform. After 3 washes in PBS at room temperature, embryos were embedded in fresh staining solution (0.5 mg/mL X-Gal, 0.25 mM K_3_Fe(CN_6_), 0.25 mM K_4_Fe(CN_6_), 0.01% NP40, 40 mM MgCl_2_ in PBS; to which 0.01% sodium deoxycholate was added for embryos ≥ E13.5) overnight at 37 °C in a rotating wheel. The enzymatic reaction was stopped with several washes in PBS, and embryos were photographed using a Leica stereo-microscope with a DF550 camera.

### 4.4. Skeletal Stainings

Alcian blue/alizarin red stainings of cartilage and bones were performed based on the protocol detailed in [30]. Briefly, embryos were collected in ice-cold PBS and stored overnight in tap water. The following morning, embryos were boiled in hot tap water for 30 s, deskinned, eviscerated and fixed in 95% ethanol for 3–5 days. Cartilage staining was performed, incubating the samples overnight in alcian blue stain (150 mg/L alcian blue 8GX; 20% glacial acetic acid in ethanol; filtered) with gentle rotation. Next, embryos were rinsed twice with 95% ethanol and extensively washed for >16 h with several changes of 95% ethanol. Embryos were cleared in 1% KOH for 10–15 min, followed by bone staining in alizarin red solution (50 mg/L alizarin red in 1% KOH; filtered) for 1–3 h. Embryos were further cleared by incubation in 1% KOH for 30 min, followed by an 80:20, 60:40, 40:60 and 20:80 1% KOH/glycerol series until destaining was complete. Dissected limbs and embryos were photographed using a Leica stereo-microscope with a DF550 camera.

### 4.5. Statistical Analyses

Unless otherwise indicated, numerical data are expressed as mean ± SEM. No statistical methods were used to predetermine sample/group sizes. Sample sizes and normalisation methods are indicated in each figure legend. Statistical analyses were performed using GraphPad Prism 8.0 software, and the corresponding method is indicated in each figure legend. The statistical significance of difference between groups is indicated by the number or asterisks (*, 0.01 < *p* < 0.05; **, 0.001 < *p* < 0.01; ***, *p* < 0.001).

## Figures and Tables

**Figure 1 ijms-23-05730-f001:**
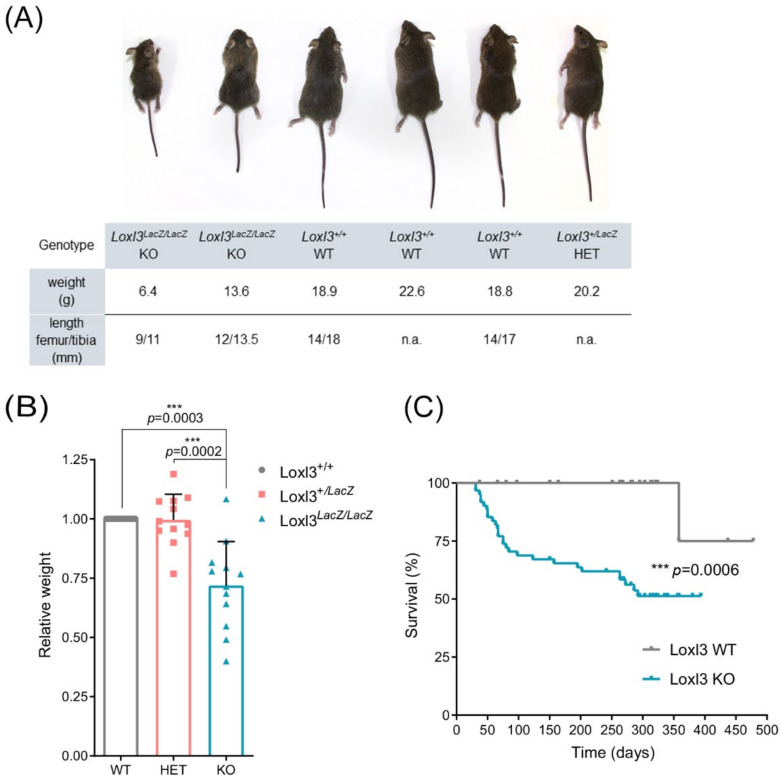
*Loxl3* KO adult mice display reduced size and poor overall survival. (**A**) Representative littermate (8 weeks old) obtained from crossing *Loxl3* heterozygous (*Loxl3^LacZ^*^/+^) mice. The weight of mice and length of femur and tibia from each mouse is depicted below. (**B**) Relative weight of wild-type, WT (*Loxl3*^+/+^), heterozygous, HET (*Loxl3*^+*/LacZ*^) and KO (*Loxl3^LacZ^*^/*LacZ*^) mice from 12 representative littermates (n = 39) from the breeding of *Loxl3* heterozygous (*Loxl3*^+*/LacZ*^) mice. Female and male mice were included, and their weight was normalised to the weight of wild-type female and male mice in each littermate. *p* values were calculated by Student’s two-tailed paired *t* test. (**C**) Kaplan–Meier survival curve of *Loxl3*^+/+^ (Loxl3 WT, n = 26) and *Loxl3^LacZ^*^/*LacZ*^ (Loxl3 KO, n = 61) mice. Perinatal death was not taken into consideration. Most adult Loxl3 KO mice died spontaneously or had to be euthanised due to their poor health status. *p* value was calculated by Mantel–Cox test.

**Figure 2 ijms-23-05730-f002:**
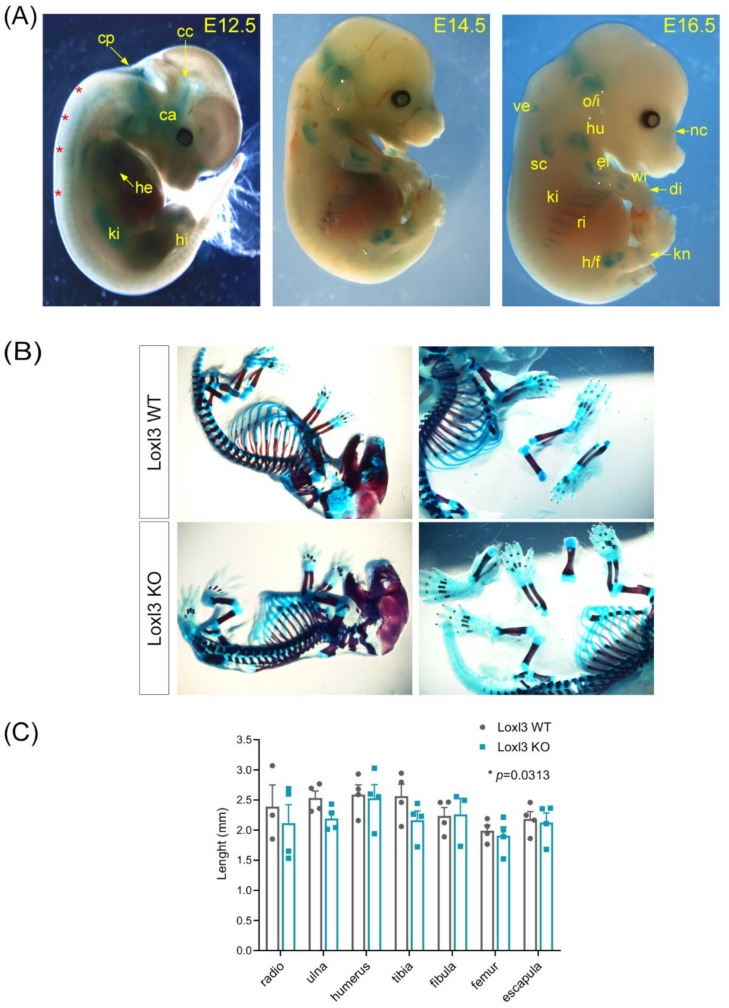
*Loxl3* expression during embryogenesis is associated with mouse skeletal development. (**A**) *Loxl3* expression pattern in embryos at indicated developmental stages as depicted by X-Gal staining. Red asterisks indicate axial staining; cp, choroid plexus; cc, chondrocranium; ca, cochlear area; he, heart; ki, kidney; hi, hindlimbs; o/i, occipital and interparietal cartilaginous precursors; ve, vertebrae; hu, humerus; nc, nasal cartilages; el, elbow; sc, scapula; wr, wrist; di, digits; ri, ribs; s/f, hip/femur; kn, knee. (**B**) Representative images of Loxl3 WT (*Loxl3*^+/+^) and KO (*Loxl3^LacZ^*^/*LacZ*^) E18.5 embryos stained with alcian blue (cartilage)/alizarin red (bone) used to measure mineralised tissues. (**C**) Length (mm) of indicated bones from Loxl3 WT (*Loxl3*^+/+^) and KO (*Loxl3^LacZ^*^/*LacZ*^) embryos (E18.5). Individual values and mean with SEM from WT (n = 4) and KO (n = 4) littermate embryos are shown. *p* value was calculated by Wilcoxon matched-pairs signed rank test.

**Figure 3 ijms-23-05730-f003:**
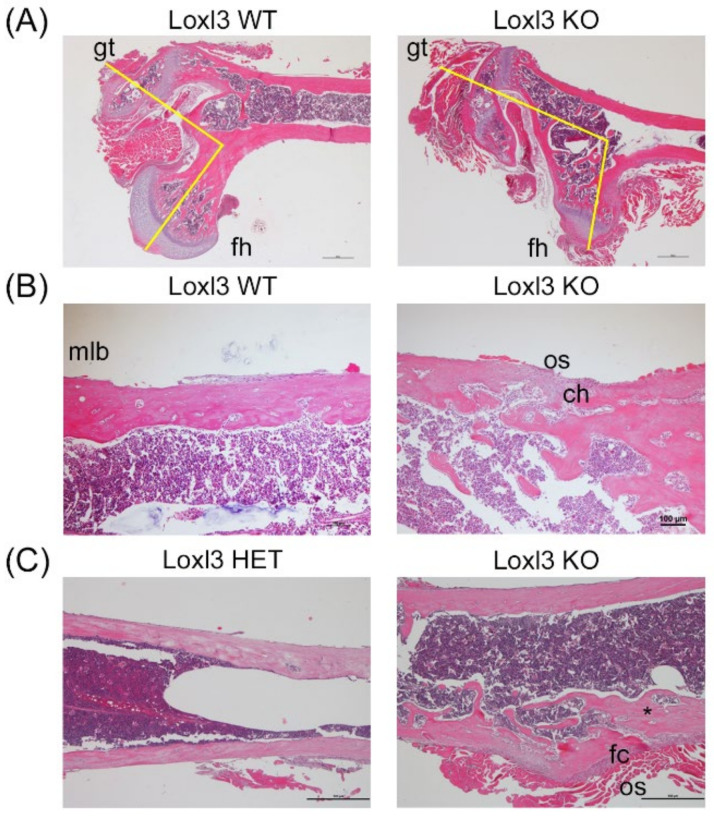
Embryonic loss of *Loxl3* promotes abnormal skeletal development. (**A**) Images showing femoral head dysplasia found in several Loxl3 KO (*Loxl3^LacZ^*^/*LacZ*^) mice with atrophy of the femoral head (fh), hypertrophy of the greater trochanter (gt) and an abnormal angle between the fh and gt, displayed by yellow lines. (**B**) The lamellar diaphysis bone in Loxl3 KO (*Loxl3^LacZ^*^/*LacZ*^) animals is interrupted by woven bone, including few chondrocytes (ch) and active osteoblasts (os) in the periphery compared to the mature lamellar bone (mlb) in WT mice. (**C**) The diaphysis from Loxl3 KO (*Loxl3^LacZ^*^/*LacZ*^) femur displays a fibro-cartilaginous (fc) area with local deformation and bone marrow disorganisation not present in a littermate heterozygous mouse (HET, *Loxl3*^+*/LacZ*^). The asterisk depicts a connective channel between the periphery and deep inter-trabecular spaces. Scale bars, 100 µm.

**Table 1 ijms-23-05730-t001:** Genotype frequency of the offspring from *Loxl3*^+/*LacZ*^ heterozygous mice crossings. Value reflecting the perinatal embryonic lethality associated with *Loxl3^LacZ^*^/*LacZ*^ genotype is indicated in bold.

Genotype	After Weaning (%)	E18.5 (%)	Expected (%)
*Loxl3* ^+/+^	30.09	18.52	25
*Loxl3* ^+/*LacZ*^	64.26	55.56	50
*Loxl3^LacZ^* ^/*LacZ*^	**5.64**	25.93	25
Total number of animals analysed	319	81	

**Table 2 ijms-23-05730-t002:** Genotype frequency of the offspring from *Loxl2*^+/−^; *Loxl3*^+/*LacZ*^ crossings. Value reflecting the postnatal lethality associated with double *Loxl2*/*Loxl3* KO is indicated in bold.

Genotype	After Weaning (%)	Expected (%)
*Loxl2*^+/+^; *Loxl3*^+/+^	13.31	6.25
*Loxl2*^+/+^; *Loxl3*^+/*LacZ*^	23.89	12.25
*Loxl2*^+/+^; *Loxl3^LacZ^*^/*LacZ*^	3.75	6.25
*Loxl2*^+/−^; *Loxl3*^+/+^	20.14	12.5
*Loxl2*^+/−^; *Loxl3*^+/*LacZ*^	34.81	25
*Loxl2*^+/−^; *Loxl3^LacZ^*^/*LacZ*^	0.68	12.5
*Loxl2*^−/−^; *Loxl3*^+/+^	2.05	6.25
*Loxl2*^−/−^; *Loxl3*^+/*LacZ*^	1.37	12.5
*Loxl2*^−/−^; *Loxl3^LacZ^*^/*LacZ*^	**0**	6.25
Total number of mice analysed	293	

**Table 3 ijms-23-05730-t003:** Genotype frequency of embryos obtained from *Loxl2*^+/−^; *Loxl3*^+/*LacZ*^ crossings. Values reflecting the embryonic lethality associated with double *Loxl2*/*Loxl3* KO are indicated in bold.

Genotype	E9.5 (%)	E11.5 (%)	E13.5 (%)	Expected (%)
*Loxl2*^+/+^; *Loxl3*^+/+^	4.82	9.38	11.11	6.25
*Loxl2*^+/+^; *Loxl3*^+/*LacZ*^	10.84	12.5	15.38	12.25
*Loxl2*^+/+^; *Loxl3^LacZ^*^/*LacZ*^	4.82	12.5	3.42	6.25
*Loxl2* ^+/−^ *; Loxl3* ^+/+^	14.46	28.13	16.24	12.5
*Loxl2*^+/−^; *Loxl3*^+/*LacZ*^	39.76	28.13	38.46	25
*Loxl2*^+/−^; *Loxl3^LacZ^*^/*LacZ*^	9.64	6.25	8.55	12.5
*Loxl2*^−/−^; *Loxl3*^+/+^	7.23	3.13	3.42	6.25
*Loxl2*^−/−^; *Loxl3*^+/*LacZ*^	3.61	0	3.42	12.5
*Loxl2*^−/−^; *Loxl3^LacZ^*^/*LacZ*^	4.82	**0**	**0**	6.25
Total number of mice analysed	83	32	117	

**Table 4 ijms-23-05730-t004:** Genotype frequency of offspring from *R26^Loxl2^*^/+^; *Loxl3*^+/*LacZ*^ crossings. Values reflecting the suppression of the embryonic lethality associated with *Loxl3* KO are indicated in bold.

Genotype	After Weaning (%)	Expected (%)
*R26^Loxl2/Loxl2^* *; Loxl3^LacZ/LacZ^*	**2**	6.25
*R26^Loxl2/+^; Loxl3^LacZ/LacZ^*	**12.67**	12.25
*R26^+/+^; Loxl3^LacZ/LacZ^*	**0**	6.25
*R26^Loxl2/Loxl2^* *; Loxl3^+/LacZ^*	5.33	12.5
*R26^Loxl2/+^* *; Loxl3^+/LacZ^*	44	25
*R26^+/+^; Loxl3^+/LacZ^*	7.33	12.5
*R26^Loxl2/Loxl2^* *; Loxl3^+/+^*	12.67	6.25
*R26^Loxl2/+^* *; Loxl3^+/+^*	14	12.5
*R26^+/+^; Loxl3^+/+^*	2	6.25
Total number of animals analysed	319	

## Data Availability

Not applicable.

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
