# Peer review of "Loxl2 and Loxl3 Paralogues Play Redundant Roles during Mouse Development"

_ijms, 2022, doi:10.3390/ijms23105730_

Round 1
Reviewer 1 Report
In this study, the authors illustrated the roles of LOXL2 and LOXL3 in mice development with more focus on skeletal abnormality. Also, with analysis of the genotype of embryos with deletion of Loxl2 and Loxl3 genes, authors suggests that both enzymes have complementary roles during mouse development which was confirmed by ubiquitous expression of Loxl2 suppression of the lethality associated with Loxl3 knockout mice. The studies in this manuscript are well-designed and conclusions are supported by data. The manuscript itself is well-written and easily readable. The couple of suggestions for the improvement of this valuable research would be:
1- In the line 133-134 the explanation of histological condition "The lamellar bone causes an attachment of cancellous bone with hematopoietic content" is a bit vague definition as the cancelous bone is normally surrounded by hematopoitic cells in bone cavity even in normal condition. It could be explained that there is more bone marrow cellularity for instance.
2- In terms of changes in endochondral ossofication between KO and wild type, one thing that would be interesting to show is epiphyseal plate in tibia.
Author Response
Reviewers´ comments
Reviewer #1
In this study, the authors illustrated the roles of LOXL2 and LOXL3 in mice development with more focus on skeletal abnormality. Also, with analysis of the genotype of embryos with deletion of Loxl2 and Loxl3 genes, authors suggest that both enzymes have complementary roles during mouse development which was confirmed by ubiquitous expression of Loxl2 suppression of the lethality associated with Loxl3 knockout mice. The studies in this manuscript are well-designed and conclusions are supported by data. The manuscript itself is well-written and easily readable.
We thank the reviewer for her/his positive comments on the findings reported in our original manuscript. We have addressed the reviewer´s concerns in the revised version and please find below (in blue) the responses to the specific questions the reviewer raised.
Specific comments:
1- In the line 133-134 the explanation of histological condition "The lamellar bone causes an attachment of cancellous bone with hematopoietic content" is a bit vague definition as the cancelous bone is normally surrounded by hematopoitic cells in bone cavity even in normal condition. It could be explained that there is more bone marrow cellularity for instance.
We agree with the remark and we have changed the sentence in lines 133-135 that now reads “In certain areas, the diaphysis compact bone is replaced by cancellous bone (Figure 3C).”
2- In terms of changes in endochondral ossofication between KO and wild type, one thing that would be interesting to show is epiphyseal plate in tibia.
Unfortunately, this cannot be analysed at this moment since we did not collect tibia from those mice when we performed our experiments. Currently that would require too much time but we will take this remark into account for future analyses to strengthen the connection between Loxl3 and endochondral ossification.

Reviewer 2 Report
The paper deal with the role of lysyl oxidase-like 2 and 3 during mouse embryogenesis. The presented data are original and are bringing new knowledge. The results are clearly presented, figure and tables are quite well arranged, discussion has no major shortcomings, references are appropriate. I suggest only minor revisions.
- Tables
- – Add (%) behind the column label – After weaning (%). E18.5 (%) etc.
- I recommend rounding the genotype frequencies of the offspring to two or three decimal places to make the result more accurate (as well as in the text).
- Lines 79/80 - Add the numerical values to describe the mortality rate more precisely.
- Figure 1C – Why does the survival curve of the Loxl3 KO mice end at day 400, whereas the survival curve of the wild type mice end at day 500?
- Lines 112/113 – Add the numerical values.
- Figure 2A – In E14.5 the Loxl3 expression pattern is not described in the figure.
- Figure 2B, legend – Add alcian blue (cartilage)/ alizarin red (bone) to the legend
- Figure 3A – The yellow lines are not visible. Please use another colour or thicker lines.
- Line 175 – Replace any with no.
- Line 176-182 – Add the numerical values.
- Line 223 – Add reference to Loxl2 mutants
- Line 231 – Or the normal expression rate of Loxl2 is insufficient and its overexpression is needed to ameliorate the Loxl3LacZ/LacZ
- Line 249 – Delete overnight
Author Response
Reviewer #2
The paper deal with the role of lysyl oxidase-like 2 and 3 during mouse embryogenesis. The presented data are original and are bringing new knowledge. The results are clearly presented, figure and tables are quite well arranged, discussion has no major shortcomings, references are appropriate. I suggest only minor revisions.
We thank the reviewer for her/his nice remarks on our work. We hope to have addressed accurately her/his concerns in the revised version of our manuscript, and provided convincing answers as detailed below (in blue).
Minor revisions:
• Tables
o – Add (%) behind the column label – After weaning (%). E18.5 (%) etc.
Done.
o I recommend rounding the genotype frequencies of the offspring to two or three decimal places to make the result more accurate (as well as in the text).
We have included two decimals as suggested in all the tables that did not include them previously (Tables 1, 2 and 4). When less than two decimals are shown is because they correspond to the value 0.
• Lines 79/80 - Add the numerical values to describe the mortality rate more precisely.
We have now included the following sentence in the text: “…determined upon closely observing Loxl3+/+ (n = 26) and Loxl3LacZ/LacZ (n = 61) mice for more than a year”.
• Figure 1C – Why does the survival curve of the Loxl3 KO mice end at day 400, whereas the survival curve of the wild type mice end at day 500?
We tried to follow a cohort of WT and KO mice for over a year. Due to space and budget constraints we could not maintain all of them alive for much longer, but we maintained a few. Those that survived longer were WT animals. The survival curve includes those animals that either died or had to be euthanized due to poor health status. At day 358, a WT mice died, but at that point there were only 3 more WT animals to supervise.
• Lines 112/113 – Add the numerical values.
Done.
• Figure 2A – In E14.5 the Loxl3 expression pattern is not described in the figure.
Indeed, we have done this on purpose. The expression patterns parallels that of E16.5 embryos and we wanted to maintain this image clean of letters in order to highlight X-Gal staining. We have now modified the sentence in the main text (line 94) to clarify this point: “At E14.5 and E16.5 stages, the expression of Loxl3 is comparable…”.
• Figure 2B, legend – Add alcian blue (cartilage)/ alizarin red (bone) to the legend
Done.
• Figure 3A – The yellow lines are not visible. Please use another colour or thicker lines.
We have increased the thickness of the lines in the corresponding panels.
• Line 175 – Replace any with no.
We have replaced “any” with “not any”, since the number of mice found were zero.
• Line 176-182 – Add the numerical values.
Done.
• Line 223 – Add reference to Loxl2 mutants.
Done.
• Line 231 – Or the normal expression rate of Loxl2 is insufficient and its overexpression is needed to ameliorate the Loxl3LacZ/LacZ
We agree, we have now included this possibility in the text in lines 232-233 as follows: “…lack of functional complementation during embryonic development is not due to alternative functions of Loxl2 and Loxl3 proteins but either to their limiting expression levels or differential temporal and/or tissue distribution during development.”
• Line 249 – Delete overnight
Done.
